# Grinding-induced supramolecular charge-transfer assemblies with switchable vapochromism toward haloalkane isomers

Jia-Rui Wu[1,2], Gengxin Wu[1], Dongxia Li[1], Meng-Hao Li[1], Yan Wang[1] & Ying-Wei Yang [1] ✉

Synthetic macrocycles have proved to be of great application value in functional charge-transfer systems in the solid state in recent years. Here we show a switchable on-off type vapochromic system toward 1-/2-bromoalkane isomers by constructing solid-state charge-transfer complexes between electron-rich perethylated pillar[5]arene and electron-deficient aromatic acceptors including 4-nitrobenzonitrile and 1,4-dinitrobenzene. These charge-transfer complexes with different colors show opposite color changes upon exposure to the vapors of 1-bromoalkanes (fading) and 2-bromoalkanes (deepening). Single-crystal structures incorporating X-ray powder diffraction and spectral analyses demonstrate that this on-off type vapochromic behavior is mainly attributed to the destruction (off) and reconstruction (on) of the charge-transfer interactions between perethylated pillar[5]arene and the acceptors, for which the competitive host-guest binding of 1-bromoalkanes and the solid-state structural transformation triggered by 2-bromoalkanes are respectively responsible. This work provides a simple colorimetric method for distinguishing positional isomers with similar physical and chemical properties.

Organic charge-transfer (CT) cocrystals have gained increasing interest in scientific communities over the past decades owing to their promising applications in, for example, ferroelectricity[1–3], (semi) conductivity[4–8], optics[9–11], magnetics[12–14] and chemical sensors[15–17]. In principle, the aromatic building blocks with opposite electric characters, that is, electron-rich π-donors (D) and electron-deficient π-acceptors (A), are responsible for specific CT/electron sharing ($\rho$) processes in organic D–A systems ($D^{+\rho}–A^{-\rho}$, $0 \le \rho \le 1$)[18–20]. The excitation wavelength of this type of resonance is usually in the visible light range, and the broad absorption bands of D–A complexes, different from the UV−vis spectra of any of its building blocks, are referred to as CT bands. Although co-crystal engineering of intermolecular D–A systems has been widely studied and applied, the design and construction of molecular cocrystals with effective CT properties is still one of the paramount challenges we face currently because of the

commonly weak CT affinities between D–A building blocks[21,22]. Therefore, exploring synergetic or assisted strategies to help enhance or stabilize intermolecular CT state is also of great significance for developing stimuli-responsive/multifunctional organic co-crystalline materials.

In supramolecular chemistry, macrocyclic receptors have been the primary tools for molecular recognition and assembly by right of their inherent cavity features and multiple noncovalent binding sites[23–32]. Taking advantage of the synergy from host−guest complexation, CT properties in macrocycles-based D–A assemblies could be robust and highly exploitable[33–54]. Pillararenes, as a relatively young generation of representative macrocyclic hosts, have promoted a phenomenal research field in modern supramolecular chemistry over the past decade and are also considered the ideal molecular entities in constructing organic CT co-crystals by virtue of their remarkable

[1]International Joint Research Laboratory of Nano-Micro Architecture Chemistry, College of Chemistry, Jilin University, 2699 Qianjin Street, 130012 Changchun, P. R. China. [2]Key Laboratory of Automobile Materials of Ministry of Education, College of Materials Science and Engineering, Jilin University, 5988 Renmin Street, 130025 Changchun, P. R. China. ✉e-mail: ywyang@jlu.edu.cn

host–guest properties, highly symmetric molecular structure, facile functionalization, and good crystallization and solubility in commonly used organic solvents[43–52]. For instance, Huang, Li, and our group reported that the pillararene-based host–guest CT cocrystals could be prepared in a modular approach, in which electron-rich pillararenes (or pillararene derivatives) could bind with various electron-deficient aromatic guests with suitable size/shape in the crystalline state through the endo-cavity and/or exo-wall CT interactions[43–46]. Moreover, Ogoshi, Huang, and Yang groups demonstrated that pillararenes themselves could simultaneously work as the donor and acceptor components in a crystalline intermolecular CT system via a facile post-modification approach[47–50]. More importantly, the marriage of CT properties into pillararene co-crystals (or crystals) provides a convenient strategy to construct vapochromic adsorption materials[51,52], in which the unoccupied (or solvent-occupied) cavity of pillararene can afford the space to capture specific organic vapors, and the vapor-induced solid-state structural transformations will change the intermolecular CT interactions between D–A pairs, thus leading to the vapochromic behaviors.

Although pioneering studies confirmed that the construction of pillararene CT co-crystals has crucial scientific value in both theory and practice; however, the research in this field is still in its infancy, and some key challenges or limitations urgently need to be licked: (i) crystalline CT complexes constructed from host–guest D–A pairs are typically obtained by cocrystallization in organic solvents, which technical process is complex, time-consuming, and environmentally unfriendly; (ii) the chromic behaviors are mainly from the slight changes of CT interactions between D–A cores upon complexing with different guest species, for which the colorimetric determination relies heavily on spectrum analysis, and can't be accurately assessed by visual inspection; (iii) the nice performance of pillararenes in selective molecular recognition and complexation has not yet been introduced into their D–A type crystalline materials.

Herein, in this work, we propose a unique competitive binding strategy to construct switchable on-off type vapochromic co-crystalline materials by the reconstruction (turn on) and destruction (turn off) of CT complexes between perethylated pillar[5]arene (EtP5) and electron-deficient aromatic guests, including 4-nitrobenzonitrile (NBN) and 1,4-dinitrobenzene (DNB) (Fig. 1a, b). Colorless EtP5 can bind with colorless NBN and brown DNB in the solid state by directly grinding their desolvated powders, respectively, and the resulting mixed powders displayed light yellow and orange-red colors as a result of the formation of CT complexes between the host-guest D–A pairs. Intriguingly, these CT interactions can be completely destroyed along with the color fading upon adsorption of the vapors of 1-bromobutane (1-BBU) and 1-bromopentane (1-BPE), while significantly enhanced with visible color deepening after exposure to the vapors of 2-bromobutane (2-BBU) and 2-bromopentane (2-BPE), thus an on-off type vapochromic system that can simultaneously distinguish and separate 1-/2-bromoalkane isomers was presented (Fig. 1c, d). Single-crystal and powder X-ray diffraction analyses revealed that the color fading and deepening are respectively attributed to the competitive host-guest binding from the 1-bromoalkane isomers, and the solid-state molecular motions to form highly ordered D–A structures triggered by the 2-bromoalkane isomers, for which the dynamic intermolecular interactions and selective molecular recognition of EtP5 jointly play a key role in determining the CT states and molecular arrangements in the resulting crystalline assemblies.

## Results
### Investigation of CT Interactions in solution
Initially, the electrostatic potential (ESP) maps of EtP5, NBN, and DNB were calculated to assess the possibility of intermolecular CT behaviors. As shown in Fig. 1a, both the "endo-cavity" and "exo-wall" of EtP5 displayed strong electronegativity owing to the electron-rich 1,4-diethoxybenzene units. By contrast, the aromatic areas of NBN and DNB were highly electropositive due to the electron-withdrawing nitrile and nitro groups (Fig. 1b). Thus, it is reasonable to expect that electron-rich EtP5 can bind with electron-deficient NBN/DNB to form host-guest CT complexes in suitable conditions.

Subsequently, the CT interactions of EtP5 toward NBN and DNB were investigated in solution. Using NBN as an example, we found that, when the colorless solutions of EtP5 (5 mM) and NBN (15 mM) were mixed in chloroform, an obvious color change occurred, and a yellow host-guest mixed solution was obtained (Fig. 2a). UV−vis spectral analysis revealed that the mixed solution showed a broad absorption band from 400 to 550 nm in the visible region, which was distinct from the spectrum of either EtP5 or NBN, indicating the existence of CT interaction and the formation of a CT complex between EtP5 and NBN. Meanwhile, similar experimental results also confirmed the CT complexation between EtP5 and DNB in chloroform solution (Fig. 2b). Besides, these host-guest CT complexations were also studied by NMR spectroscopy in CDCl$_3$ (Supplementary Figs. 1–3). Upon complexing with EtP5, the proton signals for both NBN and DNB were broadened and cannot be clearly observed in the $^1$H NMR spectra due to inclusion-induced shielding effects, which indicated that the guests were fully encapsulated in the cavity of EtP5, forming the inclusion CT complexes. Moreover, fluorescence titration experiments were carried out to quantitatively compare the CT affinities of EtP5 with NBN and DNB (Supplementary Table 1). As shown in Fig. 2c, d, the fluorescence intensity of EtP5 decreased gradually upon titration with the guests, and the binding constants were determined to be 11238 and 18086 M$^{-1}$ for NBN and DNB, respectively. As expected, EtP5 exhibited better binding affinity toward DNB than NBN, which could be attributed to the relatively weaker electron-withdrawing effect of the NBN molecule due to the existence of the nitrile group.

### Preparation of host–guest CT complexes in the solid state
Given the strong CT complexation between EtP5 and the electron-deficient guests in solution, we can reasonably speculate that this type of inclusion D–A complexes can be constructed in the solid state without the assistance of solvent crystallization. As predicted, two sets of solid-state CT complexes, referred to as EtP5-NBNα and EtP5-DNBα, were directly prepared by grinding the desolvated powder of EtP5 with NBN and DNB solids in a glass mortar, respectively. Taking EtP5-NBNα as an example, when colorless EtP5 and NBN were 1:3 (molar ratio) mixed and grinded, a significant color change occurred and a light-yellow mixture powder was yielded (Fig. 3a). Solid-state UV−vis absorption spectra presented that EtP5-NBNα has an obvious absorption band in the visible region, which is in sharp contrast to the little or no visible-range absorption of single-component EtP5 or NBN (Fig. 3b). Powder X-ray diffraction (PXRD) revealed that the phase of EtP5-NBNα differed significantly from those of EtP5, NBN, and their 1:3 (molar ratio) mixed powder without grinding, demonstrating that the mechanical grinding process led to the combination of EtP5 and NBN in the solid state and resulted in the new phase generation (Fig. 3c). Thermogravimetric analysis (TG) and differential scanning calorimetry (DSC) experiments showed that EtP5-NBNα experienced three distinct weight losses below 200 °C (Fig. 3d), wherein the first weight loss (or endothermic peak) at around 105 °C can be attributed to the loss of the unbonded NBN molecules, and the second peak at ca. 135 °C and third peak at ca. 160 °C respectively represented the loss of the bonded NBN molecules from the solid-state CT complex. It is worth noting that the weight loss of the bonded NBN can be calculated as about two molecules per EtP5, suggesting that there may be two types of host-guest binding modes coexisting in EtP5-NBNα. All the above results clearly verified the formation of a solid-state CT complex between EtP5 and NBN upon grinding the powders together. The same experimental techniques also demonstrated the formation of CT complex between EtP5 and DNB in EtP5-DNBα, as depicted in Fig. 3e–h.

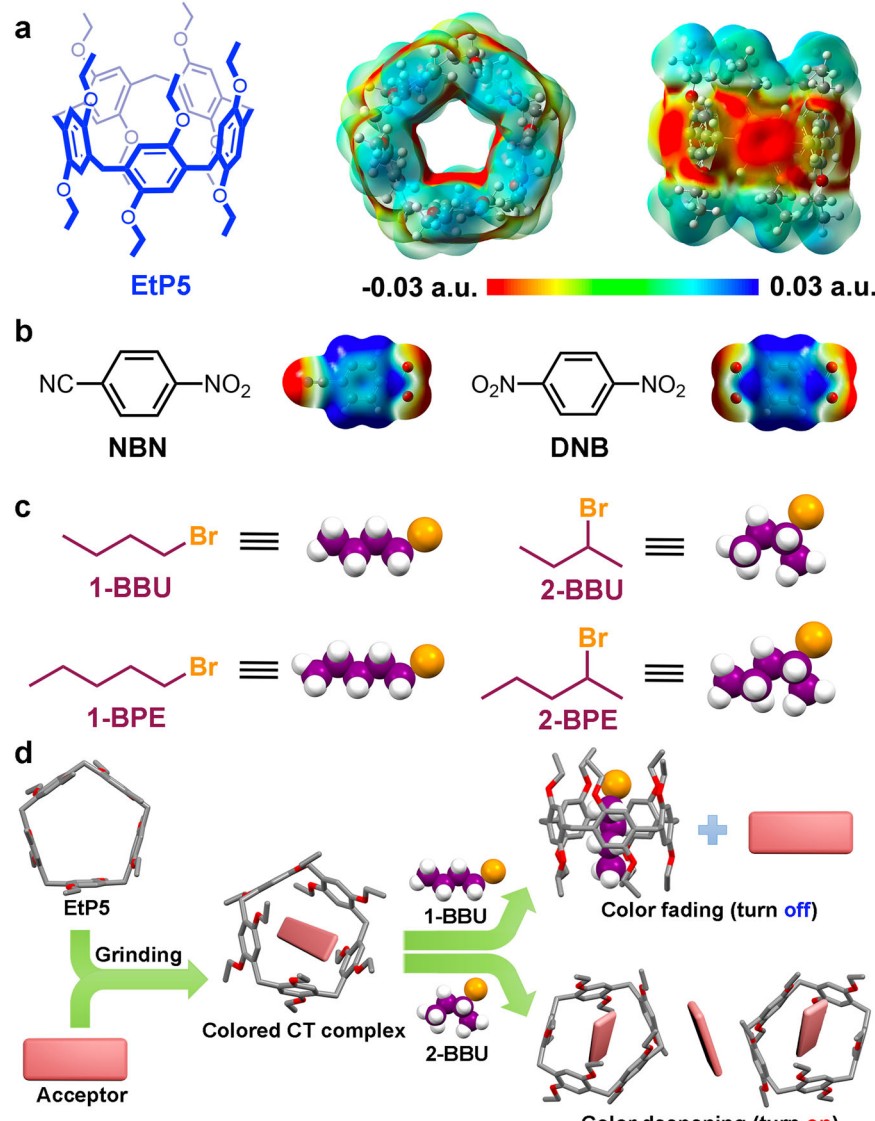

**Fig. 1 | A general strategy for constructing on-off type vapochromic system toward 1-/2-bromoalkane isomers.** Chemical structures and electrostatic potential maps of (**a**) EtP5 and (**b**) electron-deficient guests NBN and DNB. **c** Chemical structures and cartoon representations of 1-BBU, 2-BBU, 1-BPE, and 2-BPE. **d** Schematic representation of the on-off type vapochromic system in response to 1-/2-bromoalkane isomers by constructing host−guest CT assemblies based on EtP5.

## Vapochromic behaviors toward single-component bromoalkane isomer

Previously, Huang and our groups demonstrated that nonporous adaptive crystals (NACs) of pillararenes and their extended versions exhibit promising applications in the adsorption and separation of 1-/2-haloalkane isomers[55–58]. Nevertheless, examples of colorimetric sensing materials based on macrocycles for distinguishing between 1-/2-haloalkane isomers have yet to be reported so far.

On that basis, the vapochromic responses of EtP5-NBNα and EtP5-DNBα toward 1-bromoalkanes and 2-bromoalkanes were subsequently investigated through single-component solid-vapor sorption experiments. As shown in Fig. 4a, EtP5-NBNα showed an obvious color change from light-yellow to white upon exposure to the vapors of 1-BBU and 1-BPE, respectively. These color changes were determined by diffuse reflectance spectroscopy (Fig. 4b and Supplementary Fig. 14). ¹H NMR and TG-DSC results demonstrated that the 1-bromoalkane vapors were taken up by EtP5-NBNα, and the uptake amounts can be calculated as ca. 1.0 mol/mol EtP5 for both 1-BBU and 1-BPE (Supplementary Figs. 7–12). The solid-state UV−vis absorption spectra confirmed that the CT adsorption band of EtP5-NBNα entirely disappeared upon adsorption (Fig. 4c and Supplementary Fig. 13), indicating that the CT assembly between EtP5 and NBN was completely decomposed by the adsorbed 1-bromoalkanes. In sharp contrast, an obvious color deepening from light-yellow to orange was observed by exposing EtP5-NBNα to 2-BBU and 2-BPE vapors, respectively (Fig. 4a). Conversely, the CT absorption of EtP5-NBNα exhibited a distinct enhancement after exposure to the 2-bromoalkanes (Fig. 4c and Supplementary Fig. 13), which diffuse reflectance spectra also gave a clear red shift consistent with the color change (Fig. 4b and Supplementary Fig. 14). Unexpectedly, ¹H NMR and TG-DSC revealed that little 2-BBU and 2-BPE vapors were adsorbed in EtP5-NBNα (Supplementary Figs. 15–20), demonstrating that the color change was mainly attributed to the vapor-induced restructuring of EtP5-NBNα to form highly ordered host-guest CT complex. Moreover, the PXRD patterns of EtP5-NBNα after exposure to 1-/2-BPE (or 1-/2-BBU) isomers are different from each other as well as the original (Fig. 4d and Supplementary Fig. 21), further verifying the formation of two new EtP5 crystalline structures with or without CT interaction.

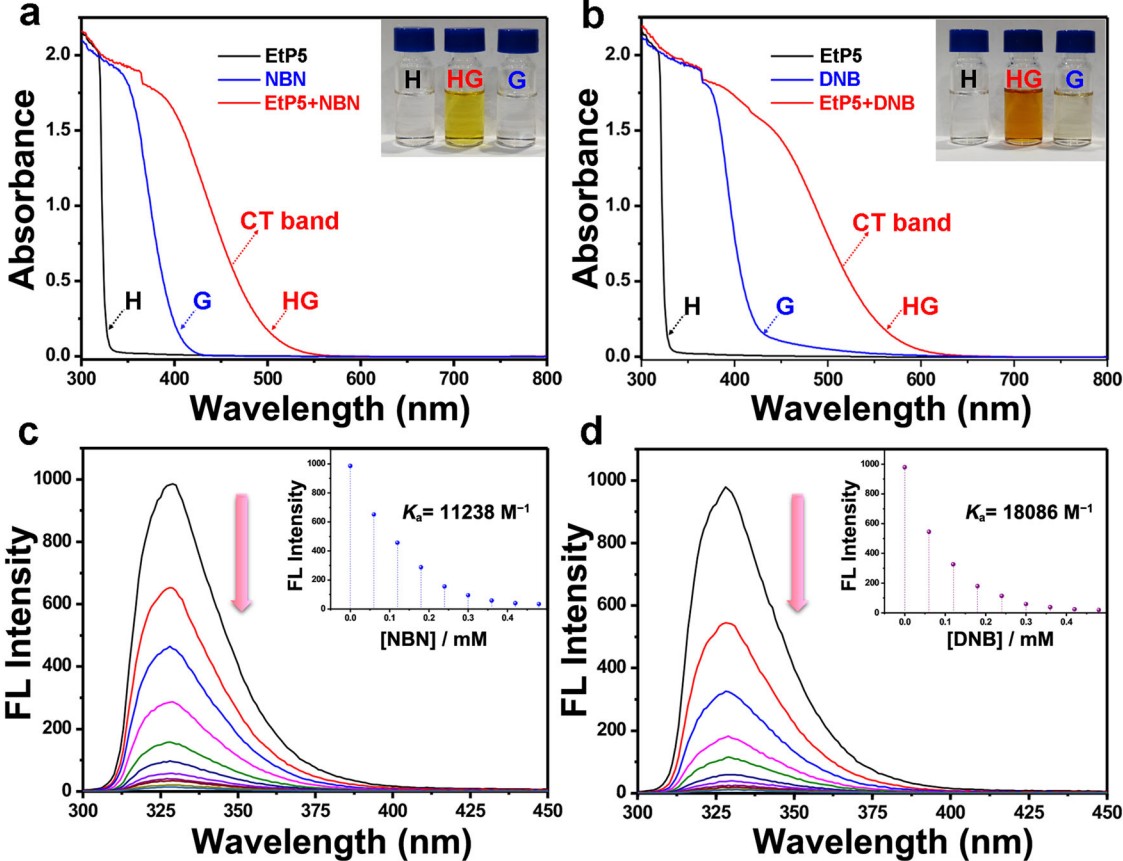

**Fig. 2 | Investigation of CT interactions in solution. a** UV–vis spectra (chloroform): (H) EtP5 (5.0 mM); (G) NBN (15 mM); (HG) EtP5 (5.0 mM) and NBN (15.0 mM). **b** UV–vis spectra (chloroform): (H) EtP5 (5.0 mM); (G) DNB (15 mM); (HG) EtP5 (5.0 mM) and DNB (15.0 mM). The inserted photographs show the solution color change due to the formation of charge-transfer assemblies between EtP6 and the corresponding guests. Fluorescence titration ($\lambda_{ex} = 280$ nm) of EtP5 (0.05 mM) in chloroform at room temperature upon titration with (**c**) NBN and (**d**) DNB (from 0 to 0.6 mM). Inset: the plots of EtP5 fluorescence intensity vs. guest concentration.

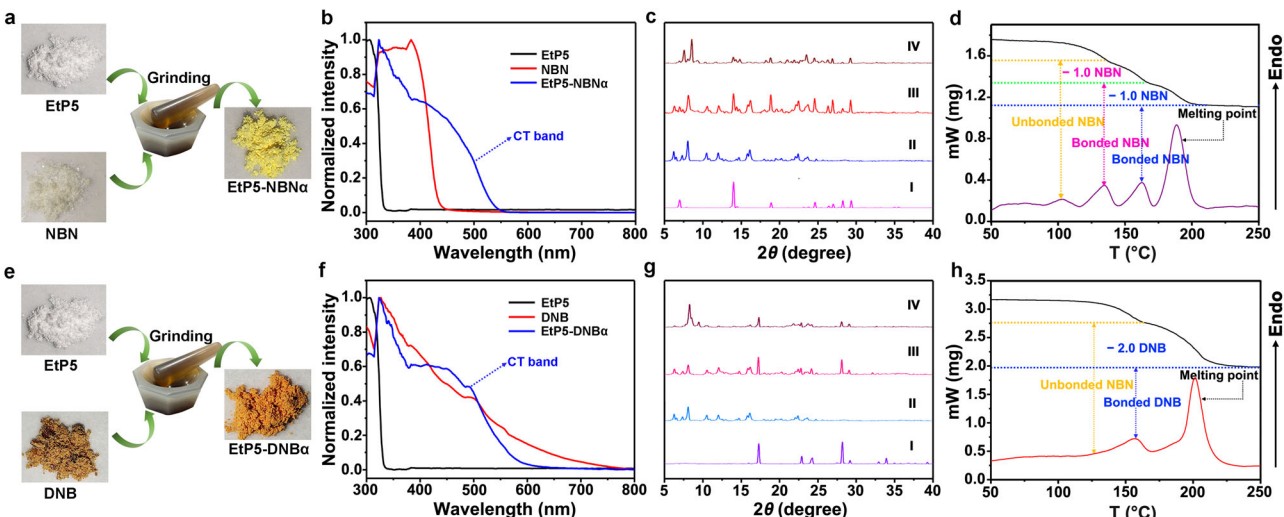

**Fig. 3 | Preparation of host–guest CT complexes in the solid state. a** Schematic representation of the procedure for preparation of EtP5-NBNα (light-yellow). Optical images show the powder color change upon forming a charge-transfer complex between EtP5 (colorless) and NBN (colorless). **b** Normalized solid-state UV–vis absorption spectra of EtP5 (black line), NBN (red line), and EtP5-NBNα (blue line). **c** PXRD patterns of (I) NBN, (II) EtP5, (III) EtP5/NBN 1:3 mixture without grinding, and (IV) EtP5-NBNα. **d** TGA (black) and DSC (violet) studies of EtP5-NBNα.

**e** Schematic representation of the procedure for preparation of EtP5-DNBα (orange-red). Optical images show the powder color change upon forming a charge-transfer complex between EtP5 (colorless) and DNB (brown). **f** Normalized solid-state UV–vis absorption spectra of EtP5 (black line), DNB (red line), and EtP5-DNBα (blue line). **g** PXRD patterns of (I) DNB, (II) EtP5, (III) EtP5/DNB 1:3 mixture without grinding, and (IV) EtP5-DNBα. **h** TGA (black) and DSC (red) studies of EtP5-DNBα.

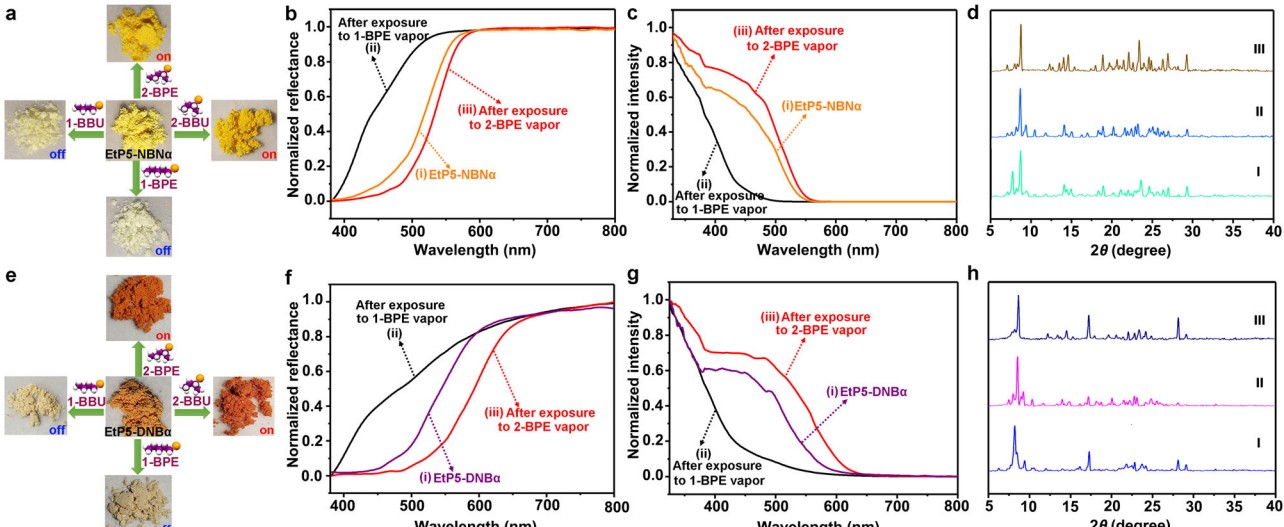

**Fig. 4 | Vapochromic behaviors toward single-component bromoalkane isomer. a** Optical images show the color changes of EtP5-NBNα upon exposure to 1-BBU, 2-BBU, 1-BPE, and 2-BPE, respectively. **b** Normalized solid-state UV−vis diffuse reflection spectra: (i) original EtP5-NBNα, after exposure to (ii) 1-BPE vapor, and (iii) 2-BPE vapor. **c** Normalized solid-state UV−vis absorption spectra: (i) original EtP5-NBNα, after exposure to (ii) 1-BPE vapor, and (iii) 2-BPE vapor. **d** PXRD patterns of EtP5-NBNα: (I) original EtP5-NBNα, after exposure to (II) 2-BPE vapor, and (III) 1-BPE vapor. **e** Optical images show the color changes of EtP5-DNBα upon exposure to 1-BBU, 2-BBU, 1-BPE and 2-BPE, respectively. **f** Normalized solid-state UV−vis diffuse reflection spectra: (i) original EtP5-DNBα, after exposure to (ii) 1-BPE vapor, and (iii) 2-BPE vapor. **g** Normalized solid-state UV−vis absorption spectra: (i) original EtP5-DNBα, after exposure to (ii) 1-BPE vapor, and (iii) 2-BPE vapor. **h** PXRD patterns of EtP5-DNBα: (I) original EtP5-DNBα, after exposure to (II) 2-BPE vapor, and (III) 1-BPE vapor.

Besides, the same experimental processes also verified the vapochromism of EtP5-DNBα toward 1-/2-bromoalkane isomers (Fig. 4e–h and Supplementary Figs. 22–36). As shown in Fig. 4e, EtP5-DNBα was orange-red and exhibited an obvious color fading to light gray after adsorption of 1-BBU and 1-BPE vapors; on the contrary, a color deepening to dark red was observed upon exposure to 2-BBU and 2-BPE vapors. All the above results clearly demonstrate that both EtP5-NBNα and EtP5-DNBα possess the capability to distinguish 1-/2-bromoalkane isomers through the approaches of destruction and reconstruction of host-guest CT interactions in the solid-state, thus leading to the on-off type vapochromic character.

**Structural analysis of EtP5 with NBN and DNB**

In order to obtain the explicit mechanism, single crystals of EtP5 complexed with NBN and DNB were subsequently prepared under different crystallization conditions. Cocrystallization of EtP5 with NBN and DNB resulted in forming three sets of host-guest CT cocrystals, referred to as EtP5-NBNβ, EtP5-DNBβ, and EtP5-DNBγ, respectively (Fig. 5). Interestingly, although no solvent molecules were discovered to participate in forming these CT cocrystals, the crystallization solvents still played a crucial role in modulating the molecular arrangement and host-guest stoichiometry in the resulting superstructures. The crystallographic data for these CT cocrystals were summarized in Supplementary Table 3.

Single crystals of EtP5-NBNβ with orange color were prepared through slow diffusion of methylcyclohexane into a solution of EtP5 and NBN in CH$_2$Cl$_2$. EtP5-NBNβ crystallized in the orthorhombic *Pbca* space group with a 2:3 binding stoichiometry between EtP5 and NBN (Fig. 5a). Interestingly, both in-cavity and exo-wall host−guest complexations were formed in this crystal superstructure. Driven by CT interaction, one NBN molecule threaded into the cavity of EtP5 and established favorable π−π interactions (A) with one of its aromatic subunits in a malposed face-to-face manner with the average plane−plane distance of 3.67 Å and corresponding dihedral angle of 2.07° (Supplementary Fig. 37). Besides, multiple C−H⋯π (B and C, distances 2.39 and 2.44 Å) and C−H⋯O (D−F, distances 2.64, 2.87, and 2.97 Å) interactions were also observed between EtP5 and the encapsulated NBN (Supplementary Fig. 38), which contributed to stabilizing the host-guest CT complex apart from π−π stacking. As for the NBN molecules located outside of the cavity, C−H⋯π (G and H, distances 2.73 and 2.73 Å) and C−H⋯O (I and J, distances 2.47 and 2.57 Å) interactions were the main exo−wall interactions (Supplementary Fig. 39), for which one uncomplexed NBN could link with two inclusion complexes to form an overall 2:3 host−guest assembly. It's also worth mentioning that the NBN molecules both inside and outside of the cavity of EtP5 are statistically disordered between two overlapped positions (Supplementary Fig. 40), and thus only the motifs with favorable occupancy were shown here for clarity. Interestingly, the terminal nitrile and nitro groups of the included NBN further interacted with the neighboring EtP5 through C−H⋯O and C−H⋯N interactions (Supplementary Fig. 41), giving rise to a staggered stacking pattern between adjacent EtP5 units along the *a* axis (Fig. 5a, right). As a result, no layer-like or tubular superstructures were observed in the packing mode of EtP5-NBNβ.

As for the CT cocrystals between EtP5 and DNB, EtP5-DNBβ with red color was obtained by the same method as EtP5-NBNβ. However, different from the orthorhombic space group and 2:3 host-guest binding ratio of EtP5-NBNβ, EtP5-DNBβ adopted the monoclinic *P2$_1$/n* space group with only one EtP5 and one DNB in the asymmetric unit (Fig. 5b). As shown in the crystal structure, each DNB molecule is fully encapsulated in the cavity of EtP5 to form in-cavity host−guest complexation, wherein multiple C−H⋯O (A-G, distances 2.67, 2.68, 2.73, 2.76, 2.77, 2.85 and 2.89 Å) and C−H⋯π interactions (H-J, distances 2.37, 2.60 and 2.66 Å) could be observed between them (Supplementary Fig. 42), helping to stabilize the inclusion CT complex. Notably, the included DNB was tilted to each aromatic subunit of EtP5, for which non-paralleled π⋯π interactions (K and L) with the distances of 3.82 Å and 3.89 Å and corresponding dihedral angles of 16.91° and 12.33° were observed (Supplementary Fig. 43). In contrast with EtP5-NBNβ, EtP5 in EtP5-DNBβ assembled into infinite 1D channels along the *a* axis with DNB molecules vertically threaded through.

CT cocrystal of EtP5-DNBγ with dark red color was obtained by slow evaporation of a 2-BPE solution of EtP5 and DNB. Similar to the case of EtP5-NBNβ, EtP5-DNBγ crystallized in the same space group

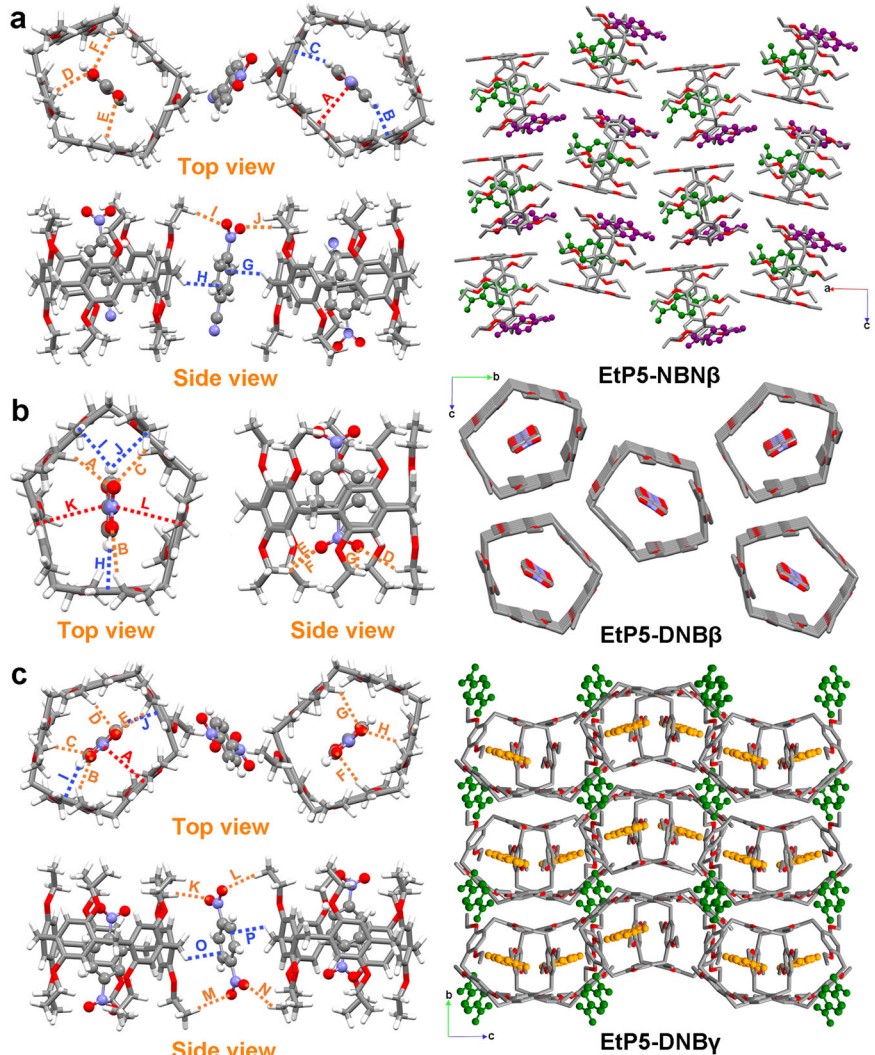

**Fig. 5 | Structural analysis of EtP5 with NBN and DNB.** X-ray single-crystal structures of (**a**) EtP5-NBNβ, (**b**) EtP5-DNBβ, and (**c**) EtP5-DNBγ. Orange, blue, and red dashed lines indicating C−H⋯O, C−H⋯π and π−π stacking interactions between EtP5 and the acceptors in these structures.

(*Pbca*, orthorhombic) and formed a 2:3 host-guest assembly with both in-cavity and exo−wall complexations (Fig. 5c and Supplementary Figs. 44–46). Different from EtP5-DNBβ, perfect face-to-face π⋯π interaction (A) with a distance of 3.66 Å and corresponding dihedral angle of 3.10° was observed between EtP5 and the encapsulated DNB in EtP5-DNBγ (Supplementary Fig. 44). Besides, diffuse reflectance spectra presented a clear red shift from EtP5-DNBβ to EtP5-DNBγ, consistent with the color deepening from red to dark red (Supplementary Fig. 47), further demonstrating the essential role of binding geometry in affecting the CT property in organic CT cocrystals.

**Mechanism study of the on-off type vapochromic behavior**
In order to clarify the mechanism of this on-off type vapochromism of EtP5-NBNα and EtP5-DNBα, 1-BBU and 1-BPE-loaded single crystals of EtP5 (referred to as 1-BBU⊂EtP5 and 1-BPE⊂EtP5) were subsequently obtained through a solution growth method and characterized by X-ray single-crystal diffraction (Supplementary Table 4). Both 1-BBU⊂EtP5 and 1-BPE⊂EtP5 adopted the orthorhombic *Pbcn* space group, wherein one linear 1-bromoalkane molecule threaded into the cavity of EtP5 to form a 1:1 host−guest complex (Fig. 6a, b). The included 1-BBU and 1-BPE molecules established favorable C−H⋯π interactions with the aromatic inner surface of EtP5, meanwhile, C−H⋯Br interactions were also observed in the cavity between them (Supplementary Figs. 48–50). Interestingly, the arrangement of EtP5 in

1-BBU⊂EtP5 and 1-BPE⊂EtP5 in a plane resulted in forming 2D layer-like superstructures with a window-to-body packing mode.

The PXRD patterns of EtP5-NBNα (or EtP5-DNBα) after adsorption of 1-BBU and 1-BPE were in good agreement with those simulated from 1-BBU⊂EtP5 and 1-BPE⊂EtP5, respectively, suggesting that the uptake of 1-BBU and 1-BPE led to the disassembly of the solid-state CT complexes and transformed the crystalline phases to 1-BBU and 1-BPE-loaded structures (Fig. 6c and Supplementary Fig. 51). Meanwhile, the PXRD patterns of EtP5-NBNα and EtP5-DNBα after exposure to 2-BBU and 2-BPE matched well with the simulated patterns from the structures of EtP5-NBNβ and EtP5-DNBγ, respectively, indicating the transformation from the initial grinded phases to ordered crystalline CT assemblies containing favorable π−π interactions and superior intermolecular CT states as aforementioned (Fig. 6d and Supplementary Fig. 52). According to the above results, we can conclude that the on-off type vapochromic property of EtP5-NBNα and EtP5-DNBα toward 1-/2-bromoalkane isomers is mainly attributed to the destruction (off) and reconstruction (on) of the CT interactions between EtP5 and the acceptors, for which the competitive host-guest binding from 1-bromoalkane isomers and the solid-state structural transformation triggered by 2-bromoalkane isomers are respectively responsible (Fig. 6e).

Interestingly, fluorescence and NMR titration experiments demonstrated that the binding constants between EtP5 and the

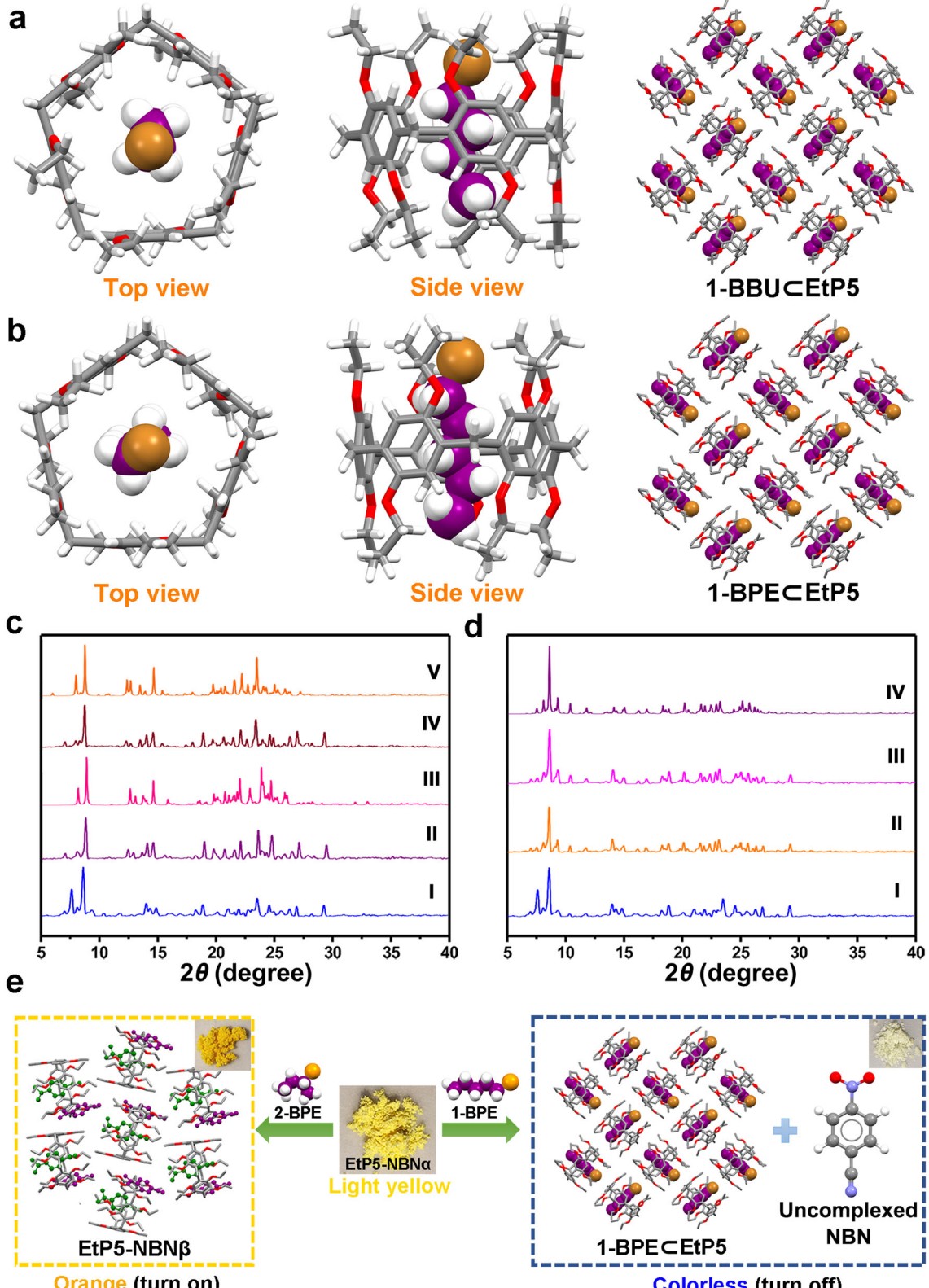

**Fig. 6 | Mechanism study of the on-off type vapochromic behavior.** X-ray single-crystal structures of (**a**) 1-BBU⊂EtP5 and (**b**) 1-BPE⊂EtP5. **c** PXRD patterns of EtP5-NBNα: (I) original EtP5-NBNα; (II) after exposure to 1-BBU vapor; (III) simulated from the crystal structure of 1-BBU⊂EtP5; (IV) after exposure to 1-BPE vapor; (V) simulated from the crystal structure of 1-BPE⊂EtP5. **d** PXRD patterns of EtP5-NBNα: (I) original EtP5-NBNα; (II) after exposure to 2-BBU vapor; (III) after exposure to 2-BPE vapor; (IV) simulated from the crystal structure of EtP5-DNBγ. **e** The mechanism of the on-off type vapochromic behavior of EtP5-NBNα in response to 1-BPE (turn off) and 2-BPE (turn on).

aromatic acceptors are much larger than that for 1-BBU and 1-BPE in solution (Supplementary Table 2), however, the guest exchange can still take place in EtP5⊃NBNα and EtP5⊃DNBα through solid–vapor phase contact. In order to clarify the mechanism for the driving force, a series of controlled experiments were subsequently carried out. First, the vapochromic behaviors of EtP5⊃NBNα and EtP5⊃DNBα toward 1-bromopropane (1-BPR) and 2-bromopropane (2-BPR) with relatively shorter alkyl chains were investigated (Supplementary Fig. 53). [1]H NMR, TG-DSC incorporating solid-state UV−vis spectral analyses revealed that the vaporized 1-BPR can't destroy the CT interactions between EtP5 and the acceptors through the guest exchange (Supplementary Figs. 54−64), which indicated the key role of effective alkyl chain lengths of 1-bromoalkanes in providing the driving force for competitive binding. Subsequently, controlled experiments by investigating the vapochromic behaviors of the crystals of EtP5⊃NBNβ and EtP5⊃DNBγ demonstrated that the guest exchanges also can't be triggered if the NBN and DNB molecules were fully encapsulated in the EtP5 cavity by the cocrystallization approach (Supplementary Figs. 65 and 66). Based on the above, we can conclude that the possibility for the guest exchange in EtP5⊃NBNα and EtP5⊃DNBα could be attributed to the relative binding capabilities of the 1-bromoalkanes toward EtP5 and the uptight/metastable complexation between EtP5 and the aromatic acceptors in the grinding state.

### Vapochromic behaviors toward the mixture of 1-/2-bromoalkane isomers

Given that EtP5⊃NBNα and EtP5⊃DNBα showed on-off type vapochromism toward 1-/2-bromoalkane isomers, we subsequently investigated whether these CT complexes possess the capability to simultaneously separate and detect 1-/2-bromoalkane isomers through solid–vapor phase contact. As shown in Fig. 7a−c, EtP5⊃NBNα exhibited obvious color fading upon exposure to 1:1 (v:v) mixtures of 1-BBU/2-BBU and 1-BPE/2-BPE, respectively, indicating that EtP5⊃NBNα still maintained its discrimination ability toward 1-bromoalkane isomers even under the mixture vapors of 1-/2-positional isomers. Consistent with the single-component adsorption, the uptake amounts of 1-BBU and 1-BPE from the mixtures were determined as nearly one mole/mole EtP5, which also gave a good agreement with the above-mentioned crystal structures (Supplementary Figs. 67 and 68). Moreover, the PXRD patterns of EtP5⊃NBNα after adsorption of the mixtures were in accordance with those of single-component 1-BBU and 1-BPE, and also matched well with the simulated patterns from 1-BBU⊂EtP5 and 1-BPE⊂EtP5, demonstrating the structural transformations from EtP5⊃NBNα to 1-BBU⊂EtP5 and 1-BPE⊂EtP5, respectively (Supplementary Figs. 69 and 70). The solid-state UV−vis absorption and diffuse reflection spectra of EtP5⊃NBNα after adsorption of the 1-BPE/2-BPE mixture vapor were in good agreement with those of single-component 1-BPE (Fig. 7b and Supplementary Fig. 71). However, the spectra of EtP5⊃NBNα after adsorption of the 1-BBU/2-BBU mixture vapor were slightly different from those of single-component 1-BBU (Fig. 7c and Supplementary Fig. 72). These spectral analysis results indicate that the competitive binding between EtP5 and 1-BBU is weaker than that between EtP5 and 1-BPE, thus leading to a restricted performance in disassembling the CT complex between EtP5 and NBN with the interference from 2-BBU molecules.

Unlike EtP5⊃NBNα, opposite color changes were observed in EtP5⊃DNBα upon exposure to the mixture vapors of 1-BBU/2-BBU and 1-BPE/2-BPE, respectively. As shown in Fig. 7d−f, EtP5⊃DNBα could distinguish and separate 1-BPE from the 1-BPE/2-BPE mixture vapor by the color fading upon 1-BPE uptake but can't keep its visual discrimination toward 1-BBU from the 1-BBU/2-BBU mixture vapor as a result of the interference from 2-BBU. [1]H NMR result revealed insufficient adsorption of 1-BBU (0.5 mol/mol EtP5) from the mixture vapor compared with the single-component 1-BBU sorption (Supplementary Fig. 73), which implied that only part of the host-guest CT complexes in EtP5⊃DNBα could be disassembled by 1-BBU with the presence of 2-BBU,

also demonstrated the key role of vapor concentration in affecting the inverted guest exchange in the metastable CT complex. Solid-state UV −vis absorption and diffuse reflection spectra also confirmed a slight CT enhancement and color deepening of EtP5⊃DNBα upon exposure to the mixture vapor due to the 2-BBU-induced structural reassembly between EtP5 and DNB (Fig. 7f and Supplementary Fig. 77). To further clarify the discrepancy, independent gradient model (IGM) analyses (visual study of weak interaction) of EtP5 with NBN, DNB, 1-BBU, and 1-BPE were carried out. The host–guest binding iso-surfaces revealed that the intermolecular interactions between EtP5 and DNB are visually stronger than those between EtP5 and NBN (Fig. 7g, h), which gave a good agreement with the above-mentioned fluorescence titration experiments, demonstrating that the opposite color changes between EtP5⊃NBNα (fading) and EtP5⊃DNBα (deepening) upon exposure to the 1-BBU/2-BBU mixture vapor are mainly attributed to the superior binding affinity between EtP5 and DNB (Supplementary Fig. 79). Analogously, NMR titration experiments and IGM analyses also verified the stronger host-guest interactions (more C−H···π interactions) in 1-BPE⊂EtP5 than that in 1-BBU⊂EtP5 (Supplementary Fig. 80), further confirming that the opposite color changes of EtP5⊃DNBα upon exposure to the mixtures of 1-BBU/2-BBU (deepening) and 1-BPE/2-BPE (fading) is derived from the differentiated competition-binding capability between 1-BPE and 1-BBU. Furthermore, we demonstrated that the removal of 1-BPE upon heating could recover the CT interactions/ assemblies in both EtP5⊃NBNα and EtP5⊃DNBα, which still maintained their turn-off vapochromic responses toward the 1-BPE/2-BPE mixture without any detectable degradation (Supplementary Figs. 81−85).

## Discussion

In summary, we have developed an intriguing on-off type vapochromic system toward 1-/2-bromoalkane isomers by the reassembly (on) and disassembly (off) of CT complexes between electron-rich EtP5 macrocycle and electron-deficient aromatic guests NBN and DNB in the solid state. We found that these crystalline host-guest CT complexes with specific colors underwent opposite color changes upon exposure to the vapors of 1-bromoalkanes (fading) and 2-bromoalkanes (deepening), respectively. X-ray single-crystal and powder diffraction, in conjunction with NMR, thermal and spectral analyses, confirmed that the on-off type vapochromic property is mainly attributed to the destruction (off) of the host-guest CT interactions between EtP5 and the acceptors by the competitive host-guest binding between EtP5 and 1-bromoalkane isomers, and the reconstruction (on) of the host-guest CT complexes to form highly ordered D−A superstructures induced by 2-bromoalkane isomers, for which the dynamic intermolecular interactions and selective molecular recognition of EtP5 synergistically determine the CT states and molecular arrangements in the resulting assemblies. Moreover, we also demonstrated that these CT complexes could simultaneously separate and detect 1-bromoalkanes from the corresponding equal volume mixtures of 1-/2-positional isomers through solid-vapor sorption. On reflection, the significance of this work can be summarized as follows: (i) solid-state CT complexes between macrocyclic host and guest species were creatively prepared by mechanically grinding without the assistance of solvent crystallization; (ii) an on-off type vapochromic system was achieved with the aid of competitive binding strategy; (iii) an unique colorimetric method for simultaneously distinguishing and separating 1-/2-bromoalkane isomers was presented. Further studies will concentrate on constructing more functional CT systems using this intriguing strategy, and other applications of the already obtained CT complexes are still under development in our laboratory.

## Methods
### Materials
Starting materials and reagents including NBN (99%), DNB (99%), 1-BBU (99%), 2-BBU (98%), 1-BPE (98%) and 2-BPE (97%) were purchased

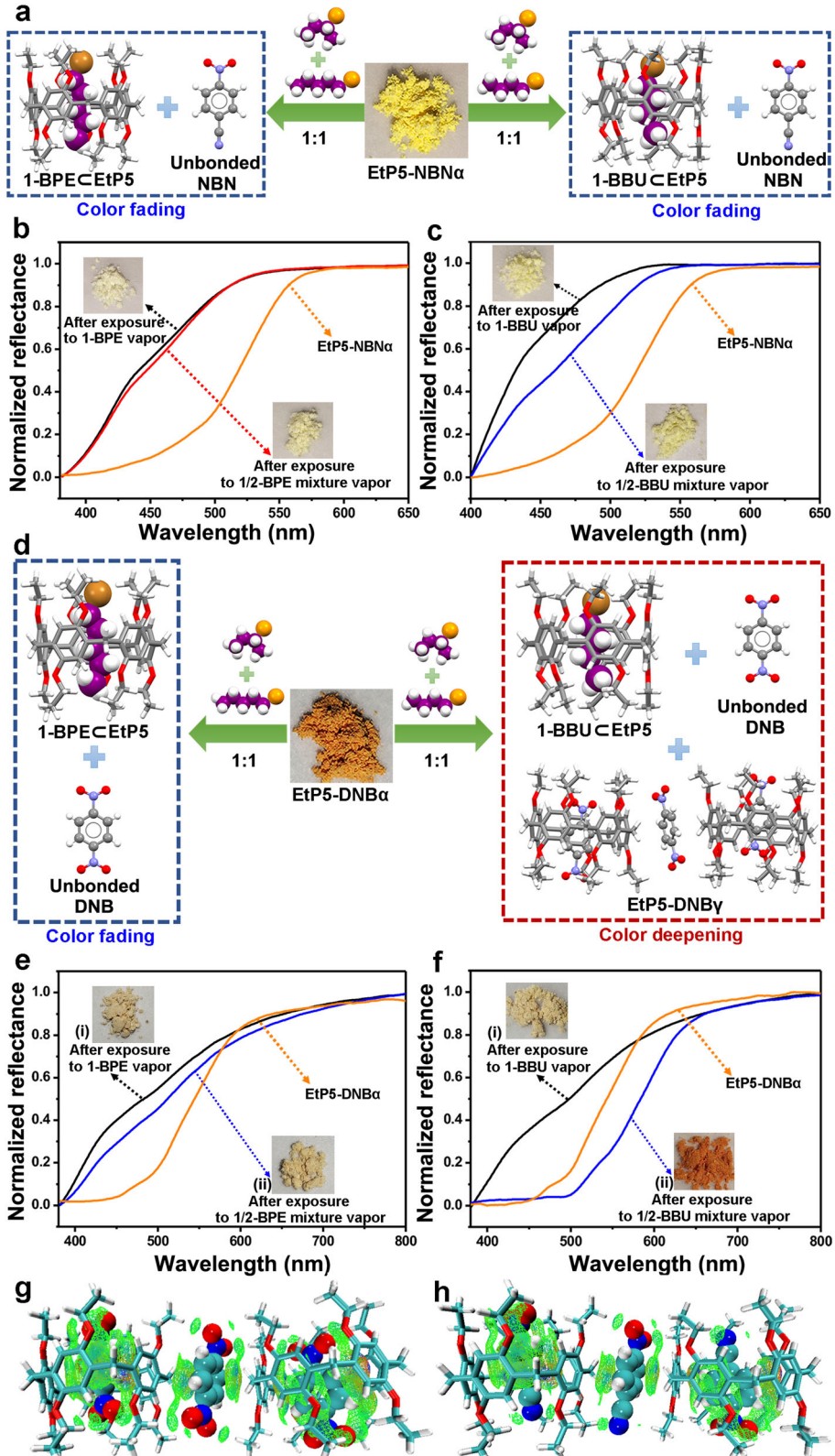

**Fig. 7 | Vapochromic behaviors toward the mixture of 1-/2-bromoalkane isomers. a** Schematic representation of the structural transformations from EtP5-NBNα to 1-bromoalkane–loaded structures upon exposure to the mixture vapors (v/v = 1:1) of 1-/2-bromoalkane isomers. Diffuse reflection spectra of (**b**) EtP5-NBNα (orange line) and after exposure to 1-BPE vapor (black line) and 1-BPE/2-BPE mixture vapor (red line), and (**c**) after exposure to 1-BBU vapor (black line) and 1-BBU/2-BBU mixture vapor (blue line). **d** Illustration of the structural transformation from EtP5- DNBα to 1-BPE⊂EtP5 upon exposure to 1-BPE/2-BPE mixture vapor, and transformation from EtP5-DNBα to a mixed phase containing EtP5-DNBγ and 1-BBU⊂EtP5 upon exposure to 1-BBU/2-BBU mixture vapor. Diffuse reflection spectra of (**e**) EtP5-DNBα (orange line) and after exposure to 1-BPE vapor (black line) and 1-BPE/2-BPE mixture vapor (blue line), and (**f**) after exposure to 1-BBU vapor (black line) and 1-BBU/2-BBU mixture vapor (blue line). Comparison of host–guest binding iso-surfaces (δginter = 0.005) between (**g**) EtP5-NBNβ and (**h**) EtP5-DNBγ.

from Energy Chemical supplier and used without further purification unless stated otherwise. Activated host–guest D–A complexes, referred to as EtP5⊃NBNα and EtP5⊃DNBα, were prepared by grinding the desolvated powder of EtP5 with NBN and DNB solids in a 1:3 molar ratio, respectively.

## Synthesis of EtP5

Compound EtP5 was prepared according to a previously reported literature[59]. 1,4-Diethoxybenzene (6.00 g, 36 mmol) and paraformaldehyde (1.08 g, 36 mmol) were added into a 100 mL round bottom flask, which was charged with dry chloroform (300 mL), and then the reaction mixture was stirred at room temperature for 20 min. Then, trifluoride diethyl etherate (4.5 mL, 36 mmol) was added to the solution. The reaction was run at room temperature for another 20 min and then quenched with NaOH solution. The organic layer was washed with water and saturated NaCl solution, and then dried with anhydrous $Na_2SO_4$. After concentrating the dried organic layer to a minimum volume, the resulting residue was subjected to silica gel column chromatograph to yield EtP5 as a white solid.

## Material characterization

$^1H$ and $^{13}C$ NMR spectra were recorded at 298 K on a Bruker AVANCEIII 400-MHz instrument at room temperature. Chemical shifts were referenced to tetramethylsilane. Powder X-ray diffraction (PXRD) measurements were collected on a PANalytical B.V. Empyrean powder diffractometer operating at 40 kV/30 mA using the Cu Kα line (λ = 1.5418 Å), and data were measured over the range 5° to 40° in 5°/min steps over 7 min. Single-crystal X-ray diffraction data were collected by a Bruker D8 Venture diffractometer equipped with a PHOTON 100 CMOS detector, using Ga-Kα radiation (λ = 1.34139 Å) and Mo-Kα radiation (λ = 0.71073 Å). Thermogravimetric analysis (TGA) and differential scanning calorimetry (DSC) experiment were carried out using a simultaneous thermal analyzer 449 F3 analyzer (NETZSCH Instruments) with an automated vertical overhead thermobalance. The samples were heated at 10 °C/min using $N_2$ as the protective gas. Fluorescent titration experiments were performed on a Shimadzu RF-5301PC spectrometer. UV-vis spectra in solution were collected on a Shimadzu UV-2550 spectrometer. Solid-state UV-vis spectra were measured by a reflectance mode on a PerkinElmer Lambda950 spectrometer from 200 to 800 nm with $BaSO_4$ as a reference.

## Vapor-phase adsorption measurements

For each solid-vapor contact experiment, an open 0.5 mL vial containing 3 mg of EtP5⊃NBNα (or EtP5⊃DNBα) was placed in a sealed 2 mL vial containing 0.01 mL of single-component bromoalkane isomer or 1:1 mixture of bromoalkane isomers. Relative uptake amounts in EtP5⊃NBNα and EtP5⊃DNBα were determined by $^1H$ NMR integrals of corresponding proton signals by completely dissolving the mixture powders in $CDCl_3$, respectively. Desorption experiments after saturation were carried out by TGA and DSC.

## Theoretical calculations

Energy-minimized structure and electrostatic potential surface (EPS) were calculated by density functional theory (DFT) using the B3LYP hybrid function combined with 6−31 G(d,p) basis set under Gaussian G09. Using single crystal superstructures as input files, independent gradient model (IGM)[60] analyses were carried out by Multiwfn 3.6 program[61] through function 20 (visual study of weak interaction) and visualized using the VMD software[62].

## Data availability

All data supporting the findings of this study are available from the article and its Supplementary Information or available from the corresponding author. The X-ray crystallographic coordinates for structures reported in this study have been deposited at the Cambridge Crystallographic Data Centre (CCDC), under deposition numbers 2247778, 2247780, 2247781, 2247790 and 2247792. These data can be obtained free of charge from The Cambridge Crystallographic Data Centre via www.ccdc.cam.ac.uk/data_request/cif.

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

## Acknowledgements
The authors gratefully acknowledge the National Natural Science Foundation of China (Grant Nos. 22201096 and 52173200), the China Postdoctoral Science Foundation (Grant No. BX2021112), and the Natural Science Foundation of Jilin Province (No. 20230101052JC) for financial support.

## Author contributions

J.-R.W. and Y.-W.Y. conceived this project and designed the experiments. J.-R.W., G.W., D.L., M.-H.L., and Y.W. conducted the experiments. All authors analyzed and interpreted the data. J.-R.W. drafted the work. Y.-W.Y. revised and finalized the manuscript. All authors approved the submitted version.

## Competing interests
The authors declare no competing interests.
