## [Peer Review File · Nature Communications]

Grinding-Induced Supramolecular Charge-Transfer Assemblies with Switchable Vapochromism Toward Haloalkane IsomersREVIEWER COMMENTS

Reviewer #1 (Remarks to the Author):

Yang and coworkers developed an unprecedented on-off type vapochromic system in this manuscript that can simultaneously distinguish and separate 1-/2-bromoalkane isomers through solid-vapor contact. To the best of our knowledge, this is the first time that solid-state CT complexes between pillararene and guests were prepared by mechanically grinding without the assistance of solvent crystallization, and also the first time that the competitive host-guest interactions were introduced to construct a vapochromic system for distinguishing isomers with slight differences in size, shape, and boiling point. Overall, this work is novel and solid, and this well-written manuscript will be of interest to the broad readership of Nature Communications. It will also bring new inspirations for host-guest/supramolecular chemistry, CT cocrystal engineering, and beyond. I highly recommend publication after minor revisions stated below:

1. Can CT behavior be observed by simply grinding the pillar[6]arene host with the guest?
2. Minor corrections: some acronyms are not bold in the main text and supplementary Information; please make them bold for the whole text.
3. Recent related literature on cocrystal-to-cocrystal transformation should be cited.

Reviewer #2 (Remarks to the Author):

In this paper formation of CT-complexes by mixing donor and acceptor units in solid state has been demonstrated and such CT-complexes exhibit on-off type vapochromic behavior which is suitable for distinguishing 1-/2-bromoalkane isomers. CT-complexes have been extensively studied in the literature including their apochromic behavior. It is not at all surprising that such complexes are formed by grinding solid donor-acceptor mixtures in the solid state and also it is not greatly advantageous as anyway once could mix them in solution and dry it. Furthermore, this is not a generally applicable tool and also selectivity toward positional isomers is not something that opens up new avenues. Overall this paper by far lacks the novelty and impact level desirable for publication in journal with IF > 10. The

paper is not recommended for publication in Nature Communication.

Reviewer #3 (Remarks to the Author):

The manuscript by Yang et al. report a competitive binding strategy to construct switchable on-off type vapochromic co-crystalline materials by the reconstruction (turn on) and destruction (turn off) of CT complexes between perethylated pillar[5]arene and electron-deficient aromatic guests, including 4-nitrobenzonitrile and 1,4-dinitrobenzene. This work is detailed and will be of interest to researchers working in the field of supramolecular chemistry and especially the researchers working on synthesis of macrocycles. The work could be of sufficient significance and may be suitable for publication in Nature Communications after major revision.

1. In page 7, the vapochromic behaviors of EtP5-NBN α and EtP5-DNB α toward single-component bromoalkane isomer had been studied via ¹H NMR and TG-DSC et al. However, in ¹H NMR spectra (in supplementary information page S5 and S6, supplementary figures 4 and 5) of EtP5-NBN α after exposure to 1-BBU or 1-BPE vapor, the signals belong to EtP5 look the same of that in ¹H NMR spectra EtP5-NBN α . It seems that the guest exchanges did not happen in solution. In the case of EtP5-DNB α , the phenomena are similar. The TGA results (in supplementary information page S6, S7, S14, and S15, supplementary figures 6, 8, 21, and 23.) of EtP5-NBN α and EtP5-DNB α after exposure to 1-BBU or 1-BPE vapor show that the lost of 1-BBU and 1-BPE, even the trapped guest NBN and DNB, are around or even lower than their boiling point (1-BPE: 130 °C; 1-BBU: 99-103 °C; NBN: 308 °C; DNB: 299.9°C). The host-guest interaction should make it harder to escape from the host cavity. Reasonable explanations should be provided in the manuscript. At the same time, it is interesting to know what happened to EtP5-NBN α and EtP5-DNB α after exposure to 1-BBU or 1-BPE vapor. And what's the driving force for the exchange between the low-boiling guest molecules and high-boiling ones.

2. In page 5, the binding constants of EtP5 toward to NBN and DNB were determined to be 11238 and 18086 M⁻¹, respectively. While, the binding constants of EtP5 toward to 1-BBU, 1-BPE, 2-BBU, and 2-BPE are not discussed in the manuscript. They should be given to better understand the guests exchange in the on-off type vapochromic system.

3. In page 13, The authors mentioned that vapochromic behaviors of EtP5-NBN α and EtP5-

DNB α toward the mixture of 1-/2-bromoalkane isomers is consistent with the single-component adsorption, and the uptake amounts of 1-BBU and 1-BPE from the mixtures were determined as nearly one mole/mole EtP5. ¹H NMR spectra of EtP5-NBN α after exposure to guest vapor were shown as evidence in supplementary information page S29 and S30, supplementary figures 50 and 51. The value of integral area of EtP5 and guests should be given.

4. In the PXRD patterns (in supplementary information page S28, S29, S33, and S34, supplementary figures 48, 49, 58 and 59) of EtP5-DNB α , the peaks of unbonded DNB crystalloids were marked via red dots. However, it is better to mark the peaks of bonded DNB guest. The bonded NBN guest also should be marked in the case of EtP5-NBN α

Reviewer#1: Yang and coworkers developed an unprecedented on-off type vapochromic system in this manuscript that can simultaneously distinguish and separate 1-/2-bromoalkane isomers through solid-vapor contact. To the best of our knowledge, this is the first time that solid-state CT complexes between pillararene and guests were prepared by mechanically grinding without the assistance of solvent crystallization, and also the first time that the competitive host-guest interactions were introduced to construct a vapochromic system for distinguishing isomers with slight differences in size, shape, and boiling point. Overall, this work is novel and solid, and this well-written manuscript will be of interest to the broad readership of Nature Communications. It will also bring new inspirations for host-guest/supramolecular chemistry, CT cocrystal engineering, and beyond. I highly recommend publication after minor revisions stated below:

Author response: We thank the reviewer for the very positive comments and the encouraging words.

(1) Can CT behavior be observed by simply grinding the pillar[6]arene host with the guest?

Author response: Thanks for this precious question. Indeed, intermolecular CT behavior can also be observed between pillar[6]arene host (EtP6) and the selected guests by simply grinding their desolvated powders due to the aromatic backbone of EtP6. As shown in Figure R1-1, when colorless EtP6 and NBN (or DNB) were mixed and grinded, a significant color change occurred and a light-yellow (or reddish brown) mixture powder was yielded. Besides, solid-state UV/Vis absorption spectra presented that the mixtures of EtP6-NBN and EtP6-DNB also showed CT absorption bands in the visible region, further confirming the formation of CT complexes in the solid state (Figure R1-1). However, cocrystallization of EtP6 with NBN and DNB would not be able to get long-range ordered cocrystal structures due to the size mismatch between the host cavity (6.7 Å) and the guests, thus restricting the application developments and mechanism analysis of the EtP6 CT complexes in this work.

Figure R1-1: (a and b) Optical images showing the powder color change upon grinding EtP6 with the guests. (c and d) Normalized solid-state UV-vis absorption spectra showing the CT bands between EtP6 and the guests upon forming the corresponding CT complexes.

(2) Minor corrections: some acronyms are not bold in the main text and supplementary Information; please make them bold for the whole text.

Author response: Thanks for pointing this out. Revised.

(3) Recent related literature on cocrystal-to-cocrystal transformation should be cited.

Author response: Added.

Reviewer#2: *In this paper formation of CT-complexes by mixing donor and acceptor units in solid state has been demonstrated and such CT-complexes exhibit on-off type vapochromic behavior which is suitable for distinguishing 1-/2-bromoalkane isomers. CT-complexes have been extensively studied in the literature including their vapochromic behavior. It is not at all surprising that such complexes are formed by grinding solid donor-acceptor mixtures in the solid state and also it is not greatly advantageous as anyway once could mix them in solution and dry it. Furthermore, this is not a generally applicable tool and also selectivity toward positional isomers is not something that opens up new avenues. Overall this paper by far lacks the novelty and impact level desirable for publication in journal with IF > 10. The paper is not recommended for publication in Nature Communication.*

Author response: Thank you for reviewing our manuscript. Indeed, it might not be a surprising strategy for the construction of CT-complexes in the solid state by directly grinding the solid donor-acceptor mixtures. However, here the authors would like to clarify and further emphasize the impact and great potential of this work. First, although the vapochromic behavior of the macrocycle-based CT-complexes has been extensively studied, however, to our best knowledge, most of the cases are derived from the modulation of the D-A binding geometry upon guest complexation, and no example of the vapochromic system based on the destruction and reconstruction of CT complexes was achieved before this work. Second, we agree that the construction of solid-state CT complexes from solution is another convenient approach. However, we firmly believe that the preparation process by dissolving and drying one by one may be more time-consuming and complex than that by grinding, not to mention the pollution problems associated with the poisonous and vaporized organic solvents. In this work, we presented a green approach. Third, the adsorption and separation of positional isomers by adaptive crystals of macrocyclic arenes is important and holds great potential in real-world applications. We believe that a kind of adsorption material with both the features of selective molecular recognition and vapochromism deserves to be appreciated and further investigated by scientists and engineers. Although there is still a long way to construct a generally applicable tool for simultaneously distinguishing and separating 1-/2-positional isomers from where we end up, pioneering studies like our contribution could be expected to stimulate chemists/physicists worldwide to provide more powerful strategies. Overall, we strongly believe that this study has potential interest to the broad readership of *Nature Communications*, especially those dedicated to solid-state host-guest chemistry and supramolecular CT materials and could also inspire researchers to continue to explore possibilities of multifunctional sensing/adsorbing materials.

Reviewer#3: *The manuscript by Yang et al. report a competitive binding strategy to construct switchable on-off type vapochromic co-crystalline materials by the reconstruction (turn on) and destruction (turn off) of CT complexes between perethylated pillar[5]arene and electron-deficient aromatic guests, including 4-nitrobenzonitrile and 1,4-dinitrobenzene. This work is detailed and will be of interest to researchers working in the field of supramolecular chemistry and especially the researchers working on synthesis of macrocycles. The work could be of sufficient significance and may be suitable for publication in Nature Communications after major revision.*

Author response: Many thanks to the reviewer for the very positive comments and the encouraging words. Here, we address the comments below.

(1) In page 7, the vapochromic behaviors of EtP5-NBNa and EtP5-DNBa toward single-component bromoalkane isomer had been studied via ¹H NMR and TG-DSC et al. However, in ¹H NMR spectra (in supplementary information page S5 and S6, supplementary figures 4 and 5) of EtP5-NBNa after exposure to 1-BBU or 1-BPE vapor, the signals belong to EtP5 look the same of that in 1H NMR spectra EtP5-NBNa. It seems that the guest exchanges did not happen in solution. In the case of EtP5-DNBa, the phenomena are similar. The TGA results (in supplementary information page S6, S7, S14, and S15, supplementary figures 6, 8, 21, and 23.) of EtP5-NBNa and EtP5-DNBa after exposure to 1-BBU or 1-BPE vapor show that the lost of 1-BBU and 1-BPE, even the trapped

guest NBN and DNB, are around or even lower than their boiling point (1-BPE: 130 °C; 1-BBU: 99-103 °C; NBN: 308 °C; DNB: 299.9°C). The host-guest interaction should make it harder to escape from the host cavity. Reasonable explanations should be provided in the manuscript. At the same time, it is interesting to know what happened to EtP5-NBN α and EtP5-DNB α after exposure to 1-BBU or 1-BPE vapor. And what's the driving force for the exchange between the low-boiling guest molecules and high-boiling ones.

Author response: Thanks for this insightful comment. To clarify the mechanism for the exchange between the low-boiling guest molecules and high-boiling ones, a series of controlled experiments were subsequently carried out. First, the vapochromic behaviors of EtP5-NBN α and EtP5-DNB α toward **1-bromopropane (1-BPR)** and **2-bromopropane (2-BPR)** with shorter alkyl chain were investigated through solid-vapor sorption experiments (Figure R3-1). Interestingly, **no color fading was observed both in EtP5-NBN α and EtP5-DNB α upon exposure to 1-BPR vapor**; Conversely, **EtP5-DNB α with orange-red color exhibited an apparent color deepening**. These color changes were determined by solid-state UV-vis absorption and diffuse reflectance spectra (Figure R3-2).

Figure R3-1. Chemical structures and cartoon representations of 1-BPR, 2-BPR, 1-BBU, 2-BBU, 1-BPE, and 2-BPE with different alkyl chain lengths.

Figure R3-2. (a) Normalized solid-state UV-vis absorption spectra: (I) original EtP5-NBN α , after exposure to (II) 1-BPR vapor, (III) 2-BPR vapor and (IV) 1-BPR/2-BPR mixture vapor. (b) Normalized diffuse reflectance spectra: (I) original EtP5-NBN α , after exposure to (II) 1-BPR vapor, (III) 2-BPR vapor and (IV) 1-BPR/2-BPR mixture vapor. (c) Normalized solid-state UV-vis absorption spectra: (I) original EtP5-DNB α , after exposure to (II) 1-BPR vapor, (III) 2-BPR vapor and (IV) 1-BPR/2-BPR mixture vapor. (d) Normalized diffuse reflectance spectra: (I) original EtP5-DNB α , after exposure to (II) 1-BPR vapor, (III) 2-BPR vapor and (IV) 1-BPR/2-BPR mixture vapor.

^1H NMR and TG-DSC revealed that few 1-BPR and 2-BPR were adsorbed in EtP5-NBN α and EtP5-DNB α , demonstrating that the vaporized **1-BPR can't destroy the CT interactions between EtP5 and the acceptors through the guest exchange** (Figures R3-3-R3-12). On this basis, we can conclude that the opposite color changes of EtP5-DNB α upon exposure to single-component 1-BPR (deepening) and 1-BBU (fading) could be ascribed to the **stronger competition-binding capability of 1-BBU than 1-BPR** toward the D-A complex (Figure R3-13). Similarly, the opposite color changes of EtP5-DNB α upon exposure to the mixtures of 1-BBU/2-BBU (deepening) and 1-BPE/2-BPE (fading) could be ascribed to the **stronger competition-binding capability of 1-BPE than 1-BBU**. Thus, there is a liner relation between the relative competition-binding capabilities of 1-bromoalkanes and the effective length of alkyl chains, for which the longer alkyl chain length provides more C-H $\cdots\pi$ interactions with the EtP5 cavity is responsible, as depicted by IGM analysis (Figure R3-14).

Figure R3-3. ^1H NMR spectra (400 MHz, CDCl_3 , 298 K) of EtP5-NBN α : (a) original EtP5-NBN α ; (b) after exposure to **1-BPR** vapor; (c) after exposure to **2-BPR** vapor.

Figure R3-4. ^1H NMR spectra (400 MHz, CDCl_3 , 298 K) of EtP5-DNB α : (a) original EtP5-DNB α ; (b) after exposure to **1-BPR** vapor; (c) after exposure to **2-BPR** vapor.

Figure R3-5. TGA of EtP5-NBN α after exposure to 1-BPR vapor.

Figure R3-6. DSC trace of EtP5-NBN α after exposure to 1-BPR vapor.

Figure R3-7. TGA of EtP5-NBN α after exposure to 2-BPR vapor.

Figure R3-8. DSC trace of EtP5-NBN α after exposure to 2-BPR vapor.

Figure R3-9. TGA of EtP5-DNB α after exposure to 1-BPR vapor.

Figure R3-10. DSC trace of EtP5-DNB α after exposure to 1-BPR vapor.

Figure R3-11. TGA of EtP5-DNB α after exposure to 2-BPR vapor.

Figure R3-12. DSC trace of EtP5-DNB α after exposure to 2-BPR vapor.

Figure R3-13. Optical images showing the color changes of (a) EtP5-NBN α and (b) EtP5-DNB α upon exposure to different bromoalkane and bromoalkane mixtures.

Figure R3-14. Host-guest binding iso-surfaces ($\delta_{\text{ginter}} = 0.005$) of (a) **1-BPR**⊂**EtP5**, (b) **1-BBU**⊂**EtP5** and (c) **1-BPE**⊂**EtP5**.

Therefore, we can conclude initially that the possibility for the guest exchange between the low-boiling guest molecules and high-boiling ones may be derived from (i) the relative binding capabilities of the 1-bromoalkanes toward EtP5; (ii) **the untight combination between EtP5 and the acceptors through the process of grinding**. Thus, controlled experiments were conducted by comparing three states of solid-state EtP5 with different binding states with the high-boiling acceptors (Figure R3-15). Concretely, pure EtP5 without NBN and DNB guests represent the completely uncomplexed/unbonded state, **EtP5-NBN α** and **EtP5-DNB α** formed by grinding the **D-A mixtures represent the intermediate or metastable combination of the D-A components**, and crystals of the D-A complexes, that is EtP5-NBN β and EtP5-DNB γ , represent the full combination of the D-A components.

Figure R3-15. Solid-state **EtP5** with different binding states with the acceptors: (a) pure EtP5; (b) **EtP5-DNB α** or **EtP5-NBN α** (grinding state); (c) **EtP5-DNB γ** or **EtP5-NBN β** (crystal state).

According to the experimental results, pure EtP5 can bind with both 1-BBU and 1-BPE without guest exchange, and metastable EtP5-NBN α and EtP5-DNB α can also bind with single-component 1-BBU and 1-BPE through the guest exchange with the bonded NBN and DNB. Interestingly, the guest exchange between 1-BBU and DNB is insufficient in EtP5-DNB α upon exposure to the mixture of 1-BBU/2-BBU, demonstrating that **the vapor concentrations of the low-boiling guests also play a key role in disassembling the metastable CT complex** apart from the differentiated binding affinity between EtP5 and the acceptors and between EtP5 and the 1-bromoalkanes as aforementioned. Interestingly, crystals of EtP5-NBN β and EtP5-DNB γ representing the most stable D-A binding state can't be disassembled by both 1-BBU and 1-BPE (Figure R3-16), which indicated that **the guest exchanges also could not be triggered if the high-boiling acceptors were fully encapsulated in the EtP5 cavity** by the cocrystallization approach, and achieve the best binding geometry/affinity with the host. Above all, we can conclude that the 1-bromoalkanes with weaker binding affinity and lower boiling points can still replace the acceptors and be encapsulated in the EtP5 cavity for the following reasons: (i) **the grinding-induced complexation between EtP5 and the acceptors is less tight and disordered in the metastable EtP5-NBN α and EtP5-DNB α** ; (ii) **the high vapor concentrations of 1-BBU and 1-BPE, could be thousands and hundreds times than the complexed NBN and DNB**, play a crucial role in driving the exchange equilibrium from high-boiling to low-boiling ones. This part of the discussion has been added to the revised manuscript.

Figure R3-16. (a) Normalized solid-state UV-vis absorption spectra: (I) EtP5-NBN β , after exposure to (II) 1-BBU vapor and (III) 1-BPE vapor. (b) Normalized solid-state UV-vis absorption spectra: (I) EtP5-DNB γ , after exposure to (II) 1-BBU vapor and (III) 1-BPE vapor. (c) PXRD patterns of EtP5-NBN β : (I) original EtP5-NBN β ; (II) after exposure to 1-BBU vapor; (III) after exposure to 1-BPE vapor; (IV) simulated from the crystal structure of EtP5-NBN β . (d) PXRD patterns of EtP5-DNB γ : (I) original EtP5-DNB γ ; (II) after exposure to 1-BBU vapor; (III) after exposure to 1-BPE vapor; (IV) simulated from the crystal structure of EtP5-DNB γ .

(2) In page 5, the binding constants of EtP5 toward to NBN and DNB were determined to be 11238 and 18086 M^{-1} , respectively. While, the binding constants of EtP5 toward to 1-BBU, 1-BPE, 2-BBU, and 2-BPE are not discussed in the manuscript. They should be given to better understand the guests exchange in the on-off type vapochromic system.

Author response: Thanks for this very valuable suggestion. The binding constants of EtP5 toward NBN and DNB were determined by fluorescence titration experiments in this work. Thus, fluorescence titration experiments were initially carried out to quantitatively compare the binding affinities of EtP5 with 1-BBU, 1-BPE, 2-BBU, and 2-BPE. As shown in Figure R3-17, the fluorescence intensity of EtP5 has no obvious decrease upon titration with the four guests, indicating the binding affinities between EtP5 and the bromoalkane isomers are too weak to afford the association constants by this method. Subsequently, the association constants for these complexes were determined using the ^1H NMR titration method (Table R3-1). As expected, the K_a value between EtP5 and 1-BPE is larger than that between EtP5 and 1-BBU due to their different alkyl chain lengths, and the K_a values for 1-BBU and 1-BPE are larger than their corresponding 2-positional isomers due to their completely different binding modes. These experimental results agreed well with the on-off type vapochromic behavior toward 1-/2-bromoalkane isomers, and this part of discussion has been added in the revised manuscript.

Figure R3-17. Fluorescence titration ($\lambda_{\text{ex}}=280$ nm) of EtP5 (0.05 mM) in chloroform at room temperature upon titration with (a) 1-BBU, (b) 2-BBU (c) 1-BPE and (d) 2-BPE (from 0 to 0.6 mM).

Table R3-1. Binding Constants (K_a) for EtP5 with 1-BBU, 1-BPE, 2-BBU, and 2-BPE at 298 K.

Guests	Solvents	K_a (M^{-1})	Binding Stoichiometry
1-BBU	CHCl ₃	$52 \pm 4^{\text{Ref R1}}$	1:1
2-BBU	CHCl ₃	a	1:1
1-BPE	CHCl ₃	70 ± 6	1:1
2-BPE	CHCl ₃	a	1:1

^aNo interactions were found or at least the association constants were too small ($<10 M^{-1}$) to be accurately calculated. Ref R1. Shu X, *et al.* Complexation of neutral 1,4-dihalobutanes with simple pillar[5]arenes that is dominated by dispersion forces. *Org. Biomol. Chem.* **10**, 3393-3397 (2012).

Figure R3-18. ^1H NMR spectra (400 MHz, CDCl_3 , 298 K) of **EtP5A** at a concentration of 1.0 mM upon addition of 2-BBU. From bottom to top, the concentration of 2-BBU was 0, 2, 4, 6, 8, 10, 12, 14, 16, 18, and 20 mM.

Figure R3-19. ^1H NMR spectra (400 MHz, CDCl_3 , 298 K) of **EtP5A** at a concentration of 1.0 mM upon addition of 1-BPE. From bottom to top, the concentration of 1-BPE was 0, 2, 4, 6, 8, 10, 12, 14, 16, 18, and 20 mM.

Figure R3-20. The non-linear curve-fitting (NMR titrations) for the complexation of EtP5A host (1.0 mM) with 1-BPE in CDCl₃ at 298 K.

Figure R3-21. ¹H NMR spectra (400 MHz, CDCl₃, 298 K) of EtP5A at a concentration of 1.0 mM upon addition of 2-BPE. From bottom to top, the concentration of 2-BPE was 0, 2, 4, 6, 8, 10, 12, 14, 16, 18 and 20 mM.

(3) In page 13, The authors mentioned that vapochromic behaviors of EtP5-NBN α and EtP5-DNB α toward the mixture of 1-/2-bromoalkane isomers is consistent with the single-component adsorption, and the uptake amounts of 1-BBU and 1-BPE from the mixtures were determined as nearly one mole/mole EtP5. ¹H NMR spectra of EtP5-NBN α after exposure to guest vapor were shown as evidence in supplementary information page S29 and S30, supplementary figures 50 and 51. The value of integral area of EtP5 and guests should be given.

Author response: Thanks for this very valuable suggestion. The requested information has been added to the revised manuscript.

(4) In the PXRD patterns (in supplementary information page S28, S29, S33, and S34, supplementary figures 48, 49, 58 and 59) of EtP5-DNB α , the peaks of unbonded DNB crystalloids were marked via red dots. However, it is better to mark the peaks of bonded DNB guest. The bonded NBN guest also should be marked in the case of EtP5-NBN α .

Author response: Thanks for this very valuable suggestion. Marked in the revised manuscript.

REVIEWERS' COMMENTS

Reviewer #1 (Remarks to the Author):

Publish as it is now.

Reviewer #3 (Remarks to the Author):

The authors have submitted a revised manuscript based on reviewers' comments. I have carefully read the latest version of the manuscript and the responding letter. The authors adequately responded to the reviewers' comments. This paper meets Nature Communications's criteria for significance and urgency and is now acceptable.

Response to Reviewers (Manuscript number: NCOMMS-23-19322A)

Reviewer#1: *Publish as it is now.*

Author response: We greatly appreciate the reviewer's efforts on helping us improve the quality of our manuscript.

Reviewer#3: *The authors have submitted a revised manuscript based on reviewers' comments. I have carefully read the latest version of the manuscript and the responding letter. The authors adequately responded to the reviewers' comments. This paper meets Nature Communications's criteria for significance and urgency and is now acceptable.*

Author response: We greatly appreciate the reviewer's efforts on helping us improve the quality of our manuscript.